# Characterizing Dissolved Organic Matter and Other Water-Soluble Compounds in Ground Ice of the Russian Arctic: A Focus on Ground Ice Classification within the Carbon Cycle Context

Petr Semenov [1,*], Anfisa Pismeniuk [2], Anna Kil [1], Elizaveta Shatrova [1,3], Natalia Belova [2], Petr Gromov [4], Sergei Malyshev [1], Wei He [5], Anastasiia Lodochnikova [1], Ilya Tarasevich [2,6], Irina Streletskaya [2] and Marina Leibman [6,7]

[1]  All-Russia Institute of Geology and Mineral Resources of the World Ocean (VNIIOkeangeologia), Saint-Petersburg 190121, Russia; a.kil@vniio.ru (A.K.); st040722@student.spbu.ru (E.S.); s.malyshev@vniio.ru (S.M.); a.lodochnikova@vniio.ru (A.L.)

[2]  Department of Cryolitology and Glaciology, Faculty of Geography, Lomonosov Moscow State University, Moscow 119991, Russia; apismeniuk@geogr.msu.ru (A.P.); belova@geogr.msu.ru (N.B.); ilya.tarasevich@student.msu.ru (I.T.); irinastrelets@geogr.msu.ru (I.S.)

[3]  Department of Geochemistry, Institute of Earth Science, Saint Petersburg State University, St. Petersburg 199034, Russia

[4]  A.P. Karpinsky Russian Geological Research Institute, St. Petersburg 199106, Russia; petr_gromov@vsegei.ru

[5]  School of Water Resources and Environment, China University of Geoscience Beijing, Beijing 100083, China; wei.he@cugb.edu.cn

[6]  Tyumen Scientific Centre SB RAS, Earth Cryosphere Institute, Tyumen 625026, Russia; m.o.lejbman@utmn.ru

[7]  Laboratory of Polar and Sub-Polar Geosystems, Institute X-BIO, University of Tyumen, Tyumen 625003, Russia

*  Correspondence: p.semenov@vniio.ru

**Abstract:** Climate-induced changes contribute to the thawing of ice-rich permafrost in the Arctic, which leads to the release of large amounts of organic carbon into the atmosphere in the form of greenhouse gases, mainly carbon dioxide and methane. Ground ice constitutes a considerable volume of the cryogenically sequestered labile dissolved organic carbon (DOC) subjected to fast mineralization upon thawing. In this work, we collected a unique geochemical database of the ground and glacier ice comprising the samples from various geographic locations in the Russian Arctic characterized by a variety of key parameters, including ion composition, carbon-bearing gases (methane and carbon dioxide), bulk biogeochemical indicators, and fluorescent dissolved organic matter (DOM) fractions. Our results show that interaction with solid material—such as sediments, detritus, and vegetation—is likely the overriding process in enrichment of the ground ice in all the dissolved compounds. Terrigenous humic-like dissolved organic matter was predominant in all the analyzed ice samples except for glacier ice from Bolshevik Island (the Severnaya Zemlya archipelago) and pure (with low sediment content) tabular ground ice from western Yamal. The labile protein-like DOM showed no correlation to humic components and was probably linked to microbial abundance in the ground ice. The sum of the fluorophores deconvoluted by PARAFAC strongly correlates to DOC, which proves the potential of using this approach for differentiation of bulk DOC into fractions with various origins and biogeochemical behaviors. The pure tabular ground ice samples exhibit the highest rate of fresh easily degradable DOM in the bulk DOC, which may be responsible for the amplification of permafrost organic matter decomposition upon thawing.

**Keywords:** ground ice; dissolved organic matter; excitation–emission matrix fluorimetry; biogeochemical cycling; greenhouse gas emission

## 1. Introduction

Climate-induced changes result in the thawing of ice-rich permafrost in the Arctic, leading to the release of significant amounts of organic carbon into the atmosphere and oceans. The mobilization of additional carbon in the modern biogeochemical cycle provokes an excess of greenhouse gases in the atmosphere, contributing to a positive feedback loop [1–4].

Ice-rich permafrost, predominantly located in Russia, Canada, and Alaska, is distinguished by widespread ground ice: constitutional ice, ice wedges, and tabular ground ice [5–9]. Thawing of ground ice in response to contemporary climate conditions leads to active layer deepening, thermokarst, thermo erosion, and slope processes, along with the release of carbon into the surrounding permafrost environments. The carbon reservoir in ground ice includes pre-formed (ice-captured) carbon-bearing greenhouse gases (GHGs, mainly $CH_4$ and $CO_2$) and pre-aged organic matter (OM), represented by dissolved (dissolved organic matter, DOM) and particulate (particulate organic matter, POM) forms. The tabular ground ice in Western Siberia has been reported to contain an enormously high amount of trapped microbial methane, which is responsible for the direct GHG emission upon the thawing of ground ice [10–12]. Although recent simulations have estimated the GHG emissions from the Yedoma deposits as only about 1% of the total permafrost GHG emissions [13], the impact of ground ice thawing on the carbon cycle is undoubtedly critical for both the local and regional ecosystems of the Arctic.

The ancient permafrost-derived organic matter has been reported to be relative enriched in biochemically labile fractions [14–16]. In this context, ground ice is particularly important. Yedoma ice wedges have been estimated to store significant pools of DOC (45.2 Tg) and dissolved inorganic carbon (DIC) (33.6 Tg) [17]. The vulnerability of ground ice DOC to modern transformation, resulting in greenhouse gas (GHG) emissions, is directly related to DOM composition. More biolabile constituents are readily mineralized, while refractory/recalcitrant constituents tend to accumulate in the environment. Incubation experiments have shown a significant increase in DOC loss in Yedoma samples after the addition of ice wedge meltwater [18,19]. This suggests that the low-molecular-weight, biolabile DOM of ice wedges promotes the mineralization of the predominantly high-molecular-weight, recalcitrant Yedoma OM through a co-metabolizing/priming effect [18,20,21]. Excitation–emission matrix (EEM) fluorescence coupled with parallel factor analysis (PARAFAC) is a widely used tool for investigating the molecular fractions of DOM composition in natural waters [22–29]. This technique allows for the calculation of fluorescent DOM components, which can be assigned to allochthonous, recalcitrant/biorefractory, or autochthonous biolabile DOM fractions. Negative correlations have been observed between DOC loss (biodegradable DOC, BDOC) and protein-like DOM content, representing fresh, autochthonous DOM in incubation tests involving ground ice and permafrost samples [18,19]. The distribution of fluorescent DOM fractions may reflect the partitioning of the DOC pool into portions with varying vulnerability to microbial transformation, ultimately resulting in different amounts and ratios of gaseous products ($CO_2$ and $CH_4$).

Fluorescent DOM fraction data can be used for paleoclimate reconstructions based on the assumption that DOM composition reflects hydrological and climate conditions as well as local vegetation of paleo environments. In our previous study, the PARAFAC-DOM component distribution in the Faddeevsky ice wedges (IWs) on Kotelny Island, New Siberian archipelago, was correlated with the age of the IW formation (Holocene and Late Pleistocene). The study also distinguished between the ice wedges and tabular ground ice within the sample collection [30].

In this study, we integrate data on water-soluble compounds in ground ice and glacier samples from various locations with significant geographic extents, including northwestern Russia (Amderma settlement), Western Siberia (the Yamal Peninsula), Eastern Siberia (Kotleny Island, the Faddeevsky Peninsula), and the Severnaya Zemlya archipelago (the Leningradsky glacier on Bolshevik Island). We address the following tasks in our work:

- Describe the variations of the available water-soluble compounds, including the carbon-bearing gases ($CH_4$ and $CO_2$) within the dataset in terms of carbon cycling.

- Compose and validate a PARAFAC model for the fluorescent DOM components in the database on ground ice from various locations in the Arctic.
- Trace the relationship between the dissolved organic carbon and the molecular fractions of DOM (PARAFAC components) and establish possible sources of the fluorescent DOM molecular fractions using available geochemical indicators.
- Estimate the water-soluble geochemical parameters' interrelation, sample variability, and intrinsic diversity using multivariate statistics (PCA).

## 2. Study Area and Dataset Design

This article presents research conducted in the Russian Arctic, based on field data collected from five distinct key sites (Figure 1). The selection of these sites was driven by their geographical location, landscape features, and the type of ice. In total, we collected samples comprising tabular ground ice (TGI) and ice wedges of various ages from north-western Russia, Western and Eastern Siberia. Furthermore, additional samples were taken from the Leningradsky glacier on Bolshevik Island in the Severnaya Zemlya archipelago for comparison with ground ice samples.

The westernmost site, located 5 km east of the Amderma settlement on the Kara Sea coast, exposes the TGI in a retrogressive thaw slump (RTS) [31]. The stratified TGI is over 4.5 m thick and is covered with loamy sediments containing fragments of mollusk shells. The stratification observed in the TGI arises from variations in the composition of soil and air inclusions within the ice. In the uppermost 1.5 m of the TGI, the sediment content is notably high, including particles of silty, sandy, and pebble sizes. Pure ice layers tend to be rather thin, typically measuring less than 1 cm in thickness. The lower part of the TGI, located at depths ranging from 6 to 9 m below the surface, represents interlayering of pure transparent ice, pure bubbly ice, and stratified ice. For this study, samples (Table 1, AM.TGI/p 1–9) were collected from depths ranging between 6 and 7 m beneath the surface.

**Table 1.** Sampling points and dataset design.

| Region | Location | Type of Ice | Age of Ice Wedges | Solid Fraction Content | Group of Samples |
|---|---|---|---|---|---|
| Yugorsky Peninsula | Amderma 69°44′53″ N 61°47′17″ E | TGI | | Pure | AM.TGI/p 1–9 |
| Yamal Peninsula | Vaskiny Dachi 70°16′03″ N 68°55′22″ E | TGI TGI IW IW | Late Pleistocene Holocene | Pure Impure | VD.TGI/p 1–2 VD.TGI/imp 1–13 VD.LPW 1–15 VD.HW 1–4 |
| | Marre-Sale 69°42′14″ N 66°48′30″ E | TGI | | Pure | MS.TGI/p 1–3 |
| New Siberian Islands | Faddeevsky, Kotelny Island 75°46′23″ N 144°8′18″ E | TGI | | Pure | F.TGI/p 1–5 |
| | 75°50′10″ N 142°47′37″ E | IW | Late Pleistocene | | F.LPW 1–8 |
| | 75°31′03″ N 145°20′49″ E | IW | Holocene | | F.HW 1–16 |
| Severnaya Zemlya | Leningradsky glacier, Bolshevik 78°24′42″ N 103°17′54″ E | Glacier | | | SZ.G1 |
| | 78°37′44″ N 104°4′36″ E | | | | SZ.G2 |
| | 78°36′21″ N 103°45′05″ E | | | | SZ.G3 |
| | 78°32′39″ N 104°54′60″ E | | | | SZ.G4 |

For box-plots, we split our dataset into 3 categories: impure TGI (TGI/imp, n = 13), pure TGI (TGI/p, n = 18), and IW (W, n = 43) by combining the distinct groups of the ground ice samples.

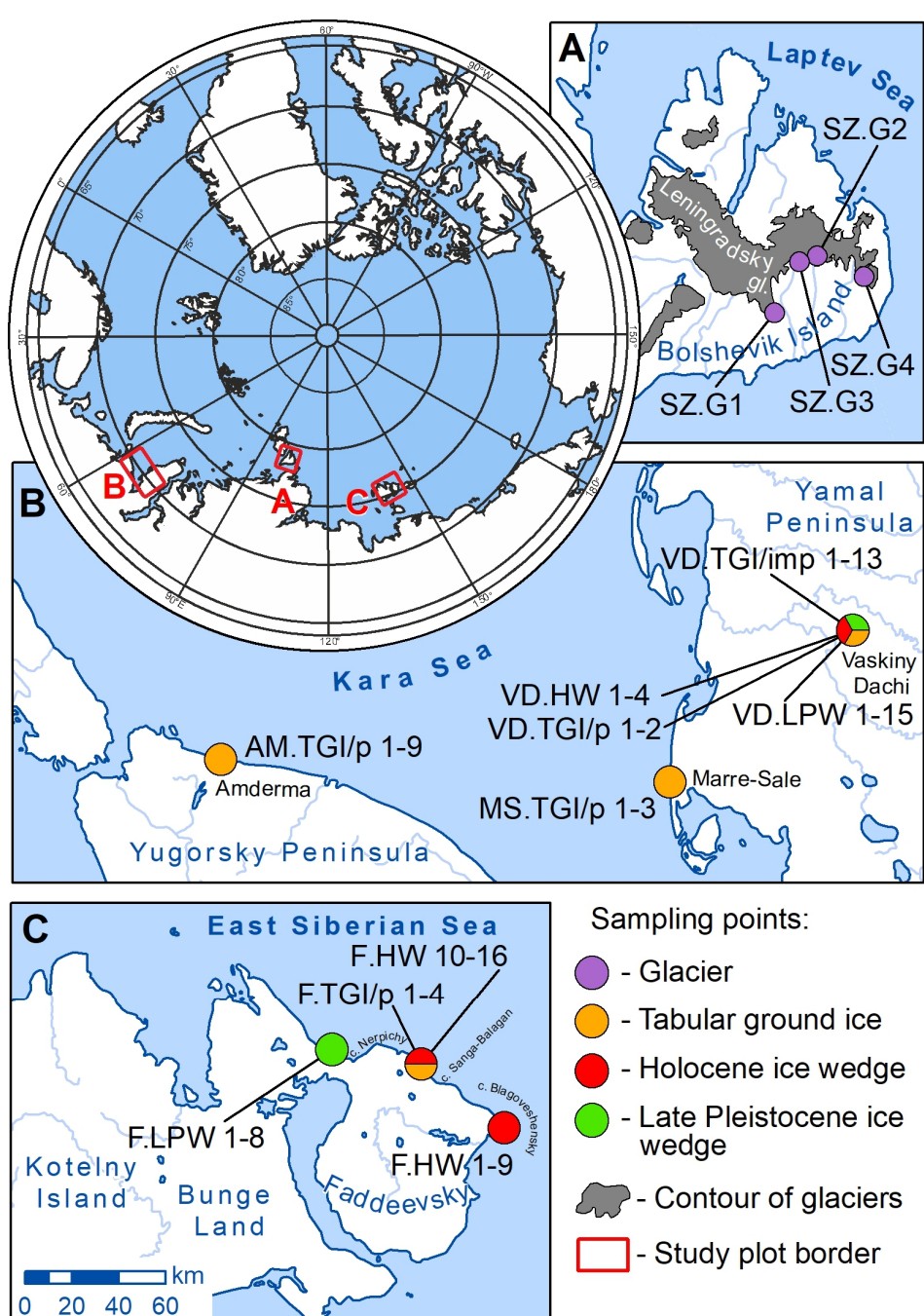

**Figure 1.** Study plots with sampling points. (**A**) Bolshevik Island, Leningradsky glacier (4 samples—SZ.G 1–4). (**B**) Yugorsky peninsula, Amderma (9 samples—AM.TGI/p 1–9); Yamal Peninsula, Marre-Sale (3 samples—MS.TGI/p 1–3), Vaskiny Dachi (34 samples—VD.TGI/imp 1–13, VD.HW 1–4, VD.TGI/p 1–2, VD.LPW 1–15). (**C**) New Siberian Islands, Kotelny Island (28 samples—F.HW 10–16, F.TGI/p 1–4, F.LPW 1–8, F.HW 1–9).

In Western Siberia, our research focused on TGI and ice wedges at Marre-Sale and Vaskiny Dachi research stations. At Marre-Sale, in 2022, we examined the TGI (TGI 1 type) [32] within a 25-m-high coastal cliff along the Kara Sea (Figure 2A). The upper part of this cliff is composed of continental sandy deposits, which contain syngenetic ice wedges. The TGI is exposed at a depth of 11–12 m from the surface and consists of two adjacent lenses, measuring 0.4 and 0.75 m in thickness. The TGI samples (Table 1, MS.TGI/p 1–3) we collected do not contain

any ground inclusions (Figure 2B), and the ice is coarse-crystalline, with individual crystals exceeding 10 cm in size, indicating a relatively slow freezing process.

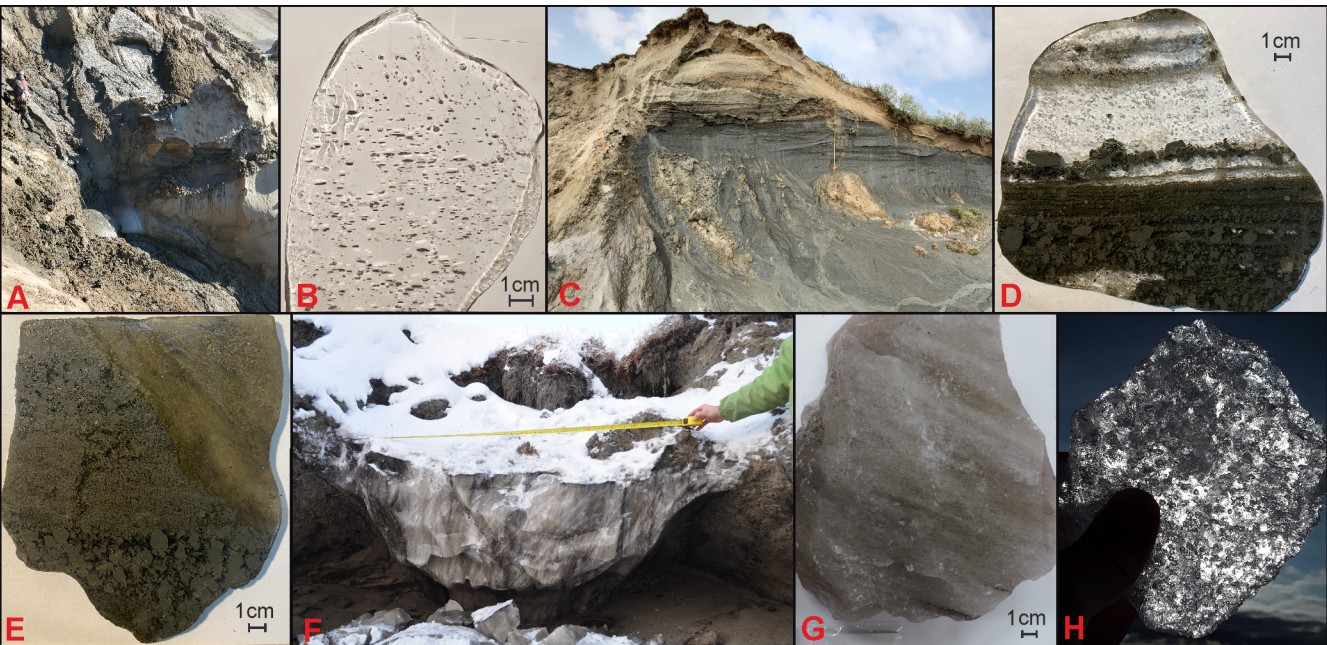

**Figure 2.** Typical examples of ground ice and glacier exposures and samples. (**A**) Tabular ground ice (TGI) in the coastal outcrop (Marre-Sale, Yamal); (**B**) pure TGI sample (Marre-Sale, Yamal); (**C**) TGI and ice wedge (IW) in the RTS of central Yamal (Vaskiny Dachi); (**D**) TGI sample (Vaskiny Dachi, Yamal); (**E**) the contact of the IW and impure TGI (Vaskiny Dachi, Yamal); (**F**) Holocene IW (Faddeevsky, New Siberian Islands); (**G**) Holocene IW sample (Faddeevsky, New Siberian Islands); (**H**) glacier ice sample (Bolshevik, Severnaya Zemlya).

At the Vaskiny Dachi Research Station, located on the watershed of the Se-Yakha and Mordy-Yakha rivers in the central part of the Yamal Peninsula, we studied TGI and IW found within a retrogressive thaw slump (RTS) (Figure 2C). The upper border of the TGI lies at a depth of 3.0 m (at an absolute height of 22 m) beneath the surface and is covered by Late Pleistocene–Holocene continental deposits. The TGI is characterized by layers of transparent ice intermixed with dark sandy loam (Figure 2D). In the middle segment of the section, the pure ice layers measure over 5 cm thick (Table 1, VD.TGI/p 1–2). The TGI is interrupted by Late Pleistocene IW (Figure 2E) with a visible height of 3.5 m and a width of 0.7 m in the upper part. We collected samples from both the IW (Table 1, VD.LPW 1–15) and the TGI (Table 1, VD.TGI/imp 1–13) throughout the entire depth of the ice wedges. Additionally, we examined Holocene ice wedges from the Vaskiny Dachi Research Station, located within the polygonal peatland. The polygons can reach sizes of up to 10 m in diameter, surrounded by polygonal troughs 0.3–0.5 m wide. Within the polygonal troughs, starting from depths of 0.5–0.6 m, we sampled the top of Holocene IW (Table 1, VD.HW 1–4). The wedge ice is transparent with distinctive sub-vertical layering and peat inclusions in lateral parts.

In the East-Siberian Arctic, we investigated ground ice on the Faddeevsky Peninsula, Kotelny Island, and the New Siberian Archipelago described in [30]. Briefly, we collected Yedoma IW samples (Table 1, F.LPW 1–8) near Cape Nerpichy from the upper part of a 12 m cliff. Close to Cape Sanga-Balagan and Blagoveshensky, we sampled Holocene ice wedges (Table 1, F.HW 1–16) found in coastal cliffs ranging from 3 to 6 m in height (Figure 2F,G). The Holocene continental complex of deposits containing ice wedges is underlain by marine clays formed before MIS 3. In proximity to Cape Sanga-Balagan, we

also investigated transparent tabular ground ice (Table 1, F.TGI/p 1–5) with some mineral inclusions at the height of 9 m. The origin of the TGI remains unclear [30].

To compare our ground ice results with glacier ice, we collected four surface samples from the southeastern part of the Leningradsky glacier, located along its southern border on Bolshevik Island in the Severnaya Zemlya archipelago. Leningradsky is a complex glacier with an approximate area of 2300 km$^2$, comprising 4 ice caps, 11 outlet glaciers, and 3 slope glaciers [33]. The age of the glacier remains uncertain. Ice core dating for the Akademii Nauk glacier suggests an age of 2.5 ka BP [34], while another study by [35] proposes an age of 12 ka BP. The Vavilova glacier is reported to have an age of 8–9 ka BP [35]. The sampling site SZ.G1 is situated in the southernmost part of shield #56 (for further reference, glacier numbers are from [33] (pp. 36–37)). The sample was obtained from the glacier's edge. Sample SZ.G2 was taken from ice cap #57, 270 m inward from the glacier's border. Sample SZ.G3 was collected from outlet glacier #68, 700 m inward from the glacier's border. Sample SZ.G4 was obtained from ice cap #59, which became separated from the complex Leningradsky glacier due to melting over the past 40 years. SZ.G4 is located in the vicinity of the glacier's highest point, approximately 2 km inward from its nearest border. All ice samples were transparent with a light-blue tint (Figure 2H), containing air bubbles, and were collected from depths of 10–15 cm.

While some of the sampling points have been previously documented by authors [30,31], this article introduces a new dataset and sampling structure based on the location and the type of ice. Sample names have been designed to convey maximum information about each sample, including its location (e.g., "SZ"—Severnaya Zemlya, Bolshevik Island; "AM"—Yugorsky Peninsula, Amderma site; "VD"—Yamal, Vaskiny Dachi site; "MS"—Yamal, Marre-Sale site; "F"—New Siberian Islands, Faddeevsky). The ice type is indicated by "G" for glacier ice, "W" for ice wedges, and "TGI" for tabular ground ice. In the case of TGI samples, the sediment content, which significantly influences DOM composition, is specified. Ice samples are classified as "pure" when the sediment content is less than 10% (by mass) of the solid fraction (TGI/p) and as "impure" when the sediment content exceeds 10% (TGI/imp). Additionally, ice wedge samples are further categorized by age into Late Pleistocene (LP) and Holocene (H). An overview of this new dataset structure is provided in Table 1. For box-plots, we split our dataset into three categories: impure TGI (TGI/imp, n = 13), pure TGI (TGI/p, n = 18), and IW (W, n = 43) by combining the distinct groups of the ground ice samples.

## 3. Materials and Methods

During the study we processed 78 ice monoliths, including glacier and ground, ice following the scheme described in the previous section. The methodology of the ice sample preparation was described in detail in our previous papers [30,36].

### 3.1. Solid Fraction Content

The ice monoliths were weighed before further preparations. After thawing of the ice monoliths at +4 °C, the supernatant melt water was taken for filtration and dissolved compounds analysis, and the precipitated solid part was freeze-dried and weighed. The weight of the coarse filter from supernatant was summed with the weight of the freeze-dried residue and related as % to the initial monolith weight.

### 3.2. Ion Composition

Ion composition of filtered thaw-water samples was analyzed using ion chromatography with a Metrohm 940 Professional IC Vario (Metrohm, Herisau, Switzerland) with a conductometry detector and a chemical suppressor unit (MSM-A). The anions were separated on a Metrosepp A Supp 5–250/4.0 column, using 5 mmoL $Na_2CO_3/NaHCO_3{}^-$ solution as an eluent with flow rate 1 mL/min. The cations were separated on a Metrosepp C6–250/4.0 column with a mixture of 1.7 mM nitric and 1.7 mM dipicolinic acid solution at a flow rate of 0.9 mL/min. The certified standard mixtures of ion composition (Fluka) were utilized to calculate the concentrations (mg/L). The uncertainty of the analytical

measurements was ±1.5%. The detection limit was 0.02 mg/L ($Cl^-$). The total dissolved ions content (TI) was determined as the sum of all ions, excluding dissolved carbonates represented as dissolved inorganic carbon (DIC).

### 3.3. Bulk Biogeochemical Parameters

Dissolved carbon species (DOC and DIC) were measured using the Shimadzu TOC-V element analyzer. The uncertainty of the analytical measurements was no higher than ±6%. Detection limit did not exceed 0.05 mgC/L. Dissolved inorganic nitrogen (DIN) was calculated as a sum of $NO_3^-$ and $NH_4^+$ related to carbon content.

### 3.4. Gas Analysis

Gas chromatographic (GC) analysis of $CH_4$ in the headspace of thaw water was carried out on a Shimadzu GC 2014 gas chromatograph (Kyoto, Japan) equipped with Restek Rt-Aluminia BOND/$Na_2SO_4$ (40 m) wide bore capillary column and flame ionization detector (FID). Helium was utilized as a carrier with 25 mL/min flow rate. For $CO_2$ concentration measurements we used a Porapak-N packed column (Shimadzu, Kyoto, Japan) (1 m) and a thermal conductivity detector (TCD—Shimadzu, Kyoto, Japan).

The certified gas mixtures were used for the method calibration. The uncertainty of the GC measurements was no higher than ±5%. The detection limits were ~0.1 ppmV for $CH_4$ and ~1 ppm for $CO_2$. Methane concentrations (ppmV) were calculated using the values of headspace mixing ratio and Bunsen solubility coefficients [37]. It must be noted that the $CO_2$ concentration in headspace gas of the meltwater samples presented in this work cannot be reliably compared to the $CO_2$ concentrations measured as result of dry ice extraction, ultimately releasing the bubble gas composition, but not including the dissolved fraction [38,39].

### 3.5. Fluorescence Measurements of Dissolved Organic Matter Molecular Composition

A Shimadzu RF5301 PC fluorimeter was applied for fluorescent excitation–emission matrix (EEM) measurements with wavelength ranges of 250–370 nm for excitation and 300–500 nm for emission of fluorescence. The fluorophores of the EEM spectra were deconvoluted by PARAFAC modeling using the Matlab graphic user interface (GUI) toolbox efc v1.2 (https://www.nomresearch.cn/efc/indexEN.html (accessed on 10 August 2023)) [40,41] and fluorescent DOM correction [41]. A dataset comprising 256 3D EEM spectra, including ice-thaw-water samples along with the natural water samples from adjacent areas (stream, lake water), was employed for PARAFAC modeling. The resulting PARAFAC model was tested through the sum of squared errors estimation (SSE), core consistency, and split-half validation procedure [42]. The relative concentration of each PARAFAC component in Raman units (RU) was acquired as Fmax output of random initialization analysis [27]. The modified Tucker's congruence coefficient (mTCC) values were employed to compare the identified PARAFAC components with library data containing 38 PARAFAC models [43].

### 3.6. Statistics

Descriptive statistics represented in this work included minimum, maximum, medium values, standard deviation, and coefficient of variation calculated for each of the 9 ground ice samples groups (Tables S1–S3). We used non-parametric Kruskal–Wallis multiple tests for comparison of ground ice sample groups to estimate the reliability of variations (at $p < 0.05$). Prior to principal component analysis (PCA), all the values in the dataset were Box–Cox transformed, and the initial and resulting data point distributions were analyzed with the Shapiro–Wilk test. The PCA was performed using 16 variables and 78 samples. The eigenvectors of the variables and the factor scores of the data were represented as a biplot. All the statistical analyses were conducted using StatSoft STATISTICA v13.5.0.17.

## 4. Results

### 4.1. Solid Fraction Content and Ion Composition of the Ground Ice

On the box-plot, representing the variance of the selected parameters between the sample categories (Figure 3A), there is an obvious separation of the impure TGI in which the ice content is minor relative to the solid one (S, %), represented either by soil or sediment particles in the thawed ground ice monoliths. A statically significant difference in solid fraction percentage is demonstrated between each of the four sample categories (G, TGI/p, TGI/imp, W) by the Kruskal–Wallis multiple test, with a p value of 0.048 for the pair TGI/p, W and $p < 0.0000$ for the pairs TGI/imp, TGI/p and TGI/imp, W. The solid fraction content (S, %) exhibits significant variability within our dataset as shown by the coefficient of variation (CV) of 189.0% (Table S1). The highest S value (68.65%) indicating the extremely high solid fraction content in the ground ice samples is detected in the impure TGI (VD.TGI/imp 9) sampled from the RTS in the Vaskiny Dachi site (central Yamal) (Figure 4A). The lowest S values, suggesting insignificant solid matter inclusions in the ice body (<0.01%), are characteristic of glacier ice and pure TGI from the Marre-Sale site (western Yamal).

The median ion concentrations as well as basic descriptive statistics for each group of samples are given in the Table S2. The most abundant ions in our sample collection (n = 78) are $Na^+$ and $Cl^-$, respectively constituting 31.85 and 42.90% of the ions (excluding $HCO_3^-$ and $CO_3^{2-}$), marked as total ion, TI (mg/L). The sequence of decreasing median values of the ion percentage for the whole dataset is: $Cl^-$ (42.90%) > $Na^+$ (31.85%) > $SO_4^{2-}$ (8.06%) > $Ca^{2+}$ (4.50%) > $K^+$ (3.48%) > $Mg^{2+}$ (3.30%) > $NH_4^+$ (1.30%) > $NO_3^{2-}$ (0.34%) > $PO_4^{3-}$ (0.12%). In this work, we use the individual ion concentrations along with their sum (TI, mg/L) in multivariate statistics analysis for better separation of the sample groups on a PCA biplot as described in Section 4.5. Thus, we do not address the particular distribution of each ion in our dataset. The ion abundance (TI, mg/L) in the samples exhibits significant variation, as indicated by a coefficient of variation (CV) of 132.08%. The distribution of TI in the categories of glacier and ground ice is illustrated by box-plots (Figure 3B). The statistically significant variance ($p = 0.0000$) is marked for TGI/imp with both TGI/p and IW.

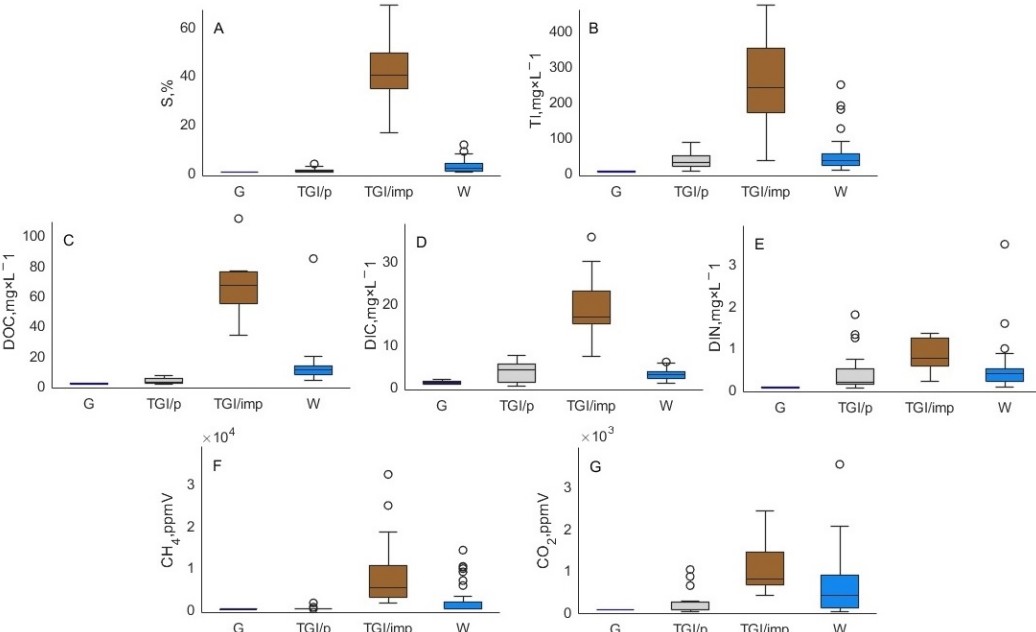

**Figure 3.** Box-plot of the solid fraction content, S (**A**), total ion content, TI (**B**), dissolved organic carbon, DOC (**C**), dissolved inorganic carbon, DIC (**D**), dissolved inorganic nitrogen, DIN (**E**), $CH_4$ (**F**), $CO_2$ (**G**) in the ground ice and glacier samples categories (G, TGI/p, TGI/imp and W).

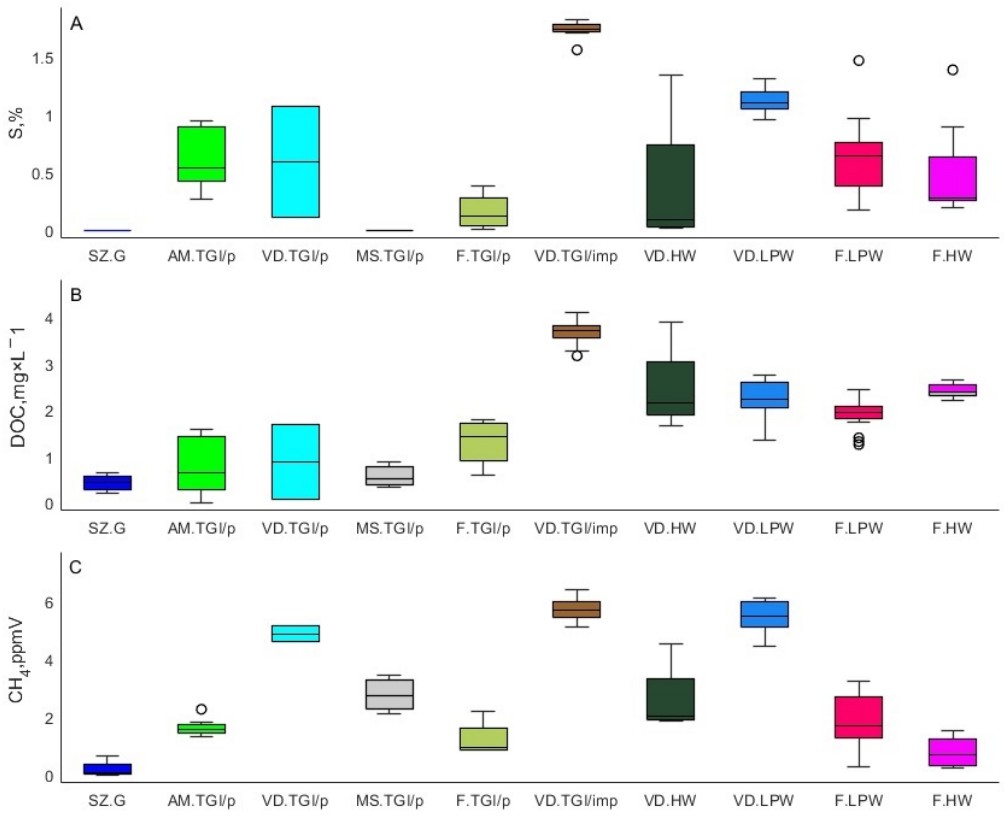

**Figure 4.** Box-plot of the parameter distribution in the individual groups of samples: solid fraction content (**A**), dissolved organic carbon (DOC) concentrations (**B**), $CH_4$ concentrations (**C**); the plotted values were Box–Cox transformed for a more complete illustration of variability.

*4.2. Bulk Dissolved Biogeochemical Parameter (DOC, DIC, DIN) Concentrations and Distribution*

Data on bulk biogeochemical parameter concentrations (DOC, DIC, DIN), including the descriptive statistics, are listed in the Table S1. Regarding DOC content, the sample categories have the following sequence of decreasing values: TGI/imp > W > TGI/p (Figure 3C) with a significant variability between the categories as shown by the non-parametric Kruskal–Wallis multiple comparison test at $p < 0.0002$. The median DOC level (66.65 mg/L) in the impure TGI samples from the Vaskiny Dachi site (VD.TGI/imp) is about 33 times higher than those of the pure TGI (TGI/p) combined group. The detected maximum of DOC concentration is as high as 111.00 mg/L in the impure TGI sample (VD.TGI/imp 11), while the minimum one is as low as 0.85 mg/L in the pure TGI from the Amderma site (AM.TGI/p 2). Variation of DOC level between the individual groups of samples naturally reveals that impure TGI samples from Yamal are sufficiently enriched in DOC relative to other groups (Figure 4B). The Holocene IWs from Vaskiny Dachi are notable for the extreme outlier (VD.HW 3) with a DOC value of 84.39 mg/L, whereas other VD.HW samples have DOC content below 10 mg/L.

Maximum DIC concentration (35.50 mg/L) is also detected in the impure TGI sample (VD.TGI/imp 11) and the minimum in the pure TGI from the Vaskiny Dachi site (VD.TGI/p 2). The median values in TGI/imp, TGI/p, and W sample categories constitute 16.73, 3.97, and 2.82 mg/L, respectively (Figure 3D). The variance between TGI/imp and other sample categories (TGI/p and W) is statistically significant at $p < 0.0002$. Maximum DIN level (3.47 mg/L) is observed in the Holocene IW from the Vaskiny Dachi site (VD.HW 3), while the minimum of 0.029 mg/L, close to the instrumental detection limit, is recorded in the pure TGI from the Amderma site (AM.TGI/p 9). The median DIN values in TGI/imp, TGI/p, and W sample categories constitute 0.73, 0.18, and 0.36 mg/L, respectively (Figure 3E). The increased level of DIN in TGI/imp samples is statistically significant, relative to both TGI/p and W ($p < 0.05$).

### 4.3. Carbon-Bearing Gas (CH₄ and CO₂) Concentrations and Distribution

Methane ($CH_4$) concentration variations are highest among the analyzed geochemical variables as indicated by the coefficient of variation (CV) that is as high as 244%. The maximum $CH_4$ concentration (32,281.65 ppmV) is detected in the impure TGI from Yamal (VD.TGI/imp 13) and the lowest (0.391 ppmV) in the Late Pleistocene IW from Faddeevsky (F.LPW 6). The corresponding box-plot diagram is represented in Figure 3F. The Kruskal–Wallis multiple test reveals a significant variance between the TGI/imp and other categories (TGI/p and W) at a level of $p < 0.001$. However, a comparison between the individual groups of the ground ice samples shows that Late Pleistocene IWs from Yamal are reliably enriched in $CH_4$, compared to the other groups ($p < 0.001$) with the single outlier. The Holocene IWs from Yamal (Vaskiny Dachi site) are remarkable for a single outlier value (418.14 ppmV) in the sample VD.HW 3, while other samples of this group have a $CH_4$ concentration below 10 ppmV.

The carbon dioxide ($CO_2$) concentrations vary from 43.120 ppmV (the minimum value) in the pure TGI of the Amderma site (AM.TGI/p 8) to a maximum of 3539 ppmV in the Holocene IW of the Vaskiny Dachi site (VD.HW 3). The box-plots of $CH_4$ variation in the individual groups are shown in Figure 4C. The box-plots of the $CH_4$ and $CO_2$ median concentrations in the categories of samples are shown in Figure 3F,G.

### 4.4. Fluorescent Dissolved Organic Matter Composition and Distribution

The employed EEM dataset is composed of 252 samples, including the ground ice (TGI, IW), glacier ice, and other natural water samples from adjacent areas (thaw-water streams and thermokarst lakes). Four fluorescence components were extracted from EEM spectra and validated for the dataset by PARAFAC analysis. Three components (P1, P2, P3) were recognized as humic-like DOM and one as protein-like (tryptophan-like DOM). In Table 2, the spectral characteristics of the extracted fluorophores are indicated as well as the results of the library search using the EFC software v1.2. The representative EEM matrices illustrating the typical EEM fingerprints of the different ground ice types are shown in Figure 5. Among the obtained components, the humic ones (P1, P2, P3, P4) correlate quite well to each other (as indicated by the Pearson's correlation values given in Table S4), but the protein-like component (P4) was equally unrelated to any of the other fractions.

**Table 2.** Spectral characteristic of the extracted PARAFAC compounds including results of the comparison with previous studies through EEM library search (EFC 3.1 software). The mTCC is modified Tucker's congruence coefficient values [27,41].

| Component | Emission Maxima | Excitation, Max | Description | Comparison with Previous Study (Library Search) |
|---|---|---|---|---|
| P1 | 470 | 370 | Humic-like | mTCC = 0.97; humic-like, but not quinone-like [44] |
| P2 | 425 | 310 | Humic-like | mTCC = 0.99, terrestrial or ubiquitous humic-like components, a photoproduct or a photorefractory component [45] |
| P3 | 470 | 260 | Humic-like | mTCC = 0.97, humic-like, UV and visible, terrestrial, a biorefractory component [46] |
| P4 | 340 | 270 | Protein-like | mTCC = 0.90. tryptophan-like, more biodegradable [45] |

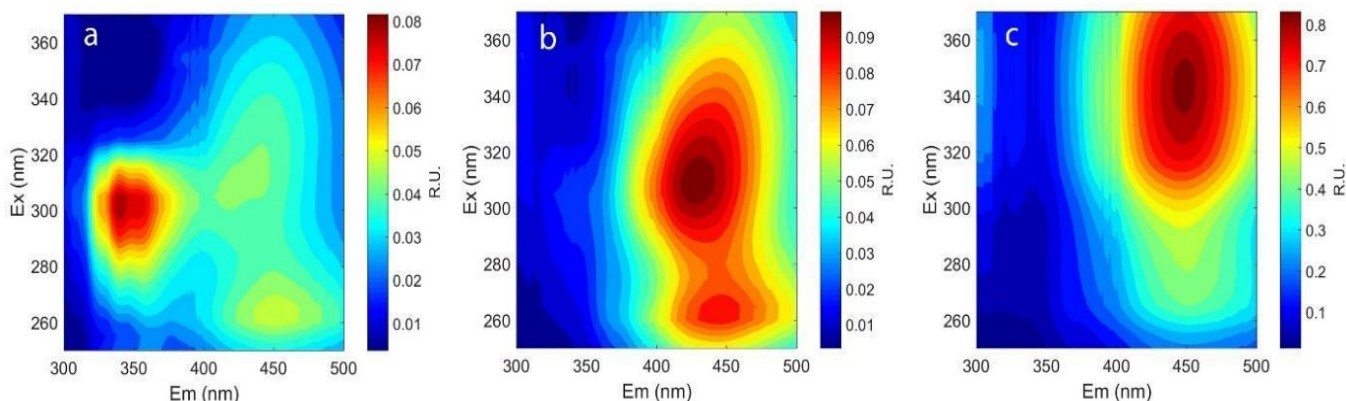

**Figure 5.** Representative EEM PARAFAC fluorograms of the various samples (pure TGI, obtained through random initialization of the 4-component PARAFAC model. (**a**) Pure TGI (AM.TGI/p 7); (**b**) Late Pleistocene IW (VD.LPW); (**c**) impure TGI (VD.TGI/imp).

The relative abundances of the EEM PARAFAC compounds (%) based on their median concentrations (RU) clearly indicate the character of the fluorescent DOM composition intergroup variability (Figure 6). Thus, the protein-like component P4 makes up >88% of the net DOM in the samples of Leningradsky glacier, Bolshevik Island (SZ.G) DOM; >81% of the pure TGI DOM from Marre-Sale (MS.TGI/p); >46% of the pure TGI from the Faddeevsky Peninsula (New Siberian Island, F.TGI/p); >15% of the pure TGI from the Yugorsky Peninsula (Amderma site, AM.TGI/p). In the other groups of analyzed samples, the contribution of P4 varies from 11.53% in the Holocene IWs of Faddeevsky (F.HW) to the minimum value of 1.1% in the Holocene IW from the Vaskiny Dachi site. Among the humic-like DOM, the component P1 is the most ubiquitous in the majority of the sample groups, excluding the SZ.G and MS.TGI/p, reported for outstanding abundance of P4. The maximum rate of P1 (83.2%) is characteristic of the impure TGI from central Yamal (VD.TGI/imp). The P2 DOM is the minor humic-like species with contribution ranging from 1.73% in the samples of Leningradsky glacier, Bolshevik Island to 6.03% in the pure TGI of the Marre-Sale site (MS.TGI/p). The component P3 is second in abundance after P1 reaching its maximum (33%) in the Late Pleistocene IW of Faddeevsky (Kotelny, New Siberian Islands), while it drops to 0.3% in SZ.G.

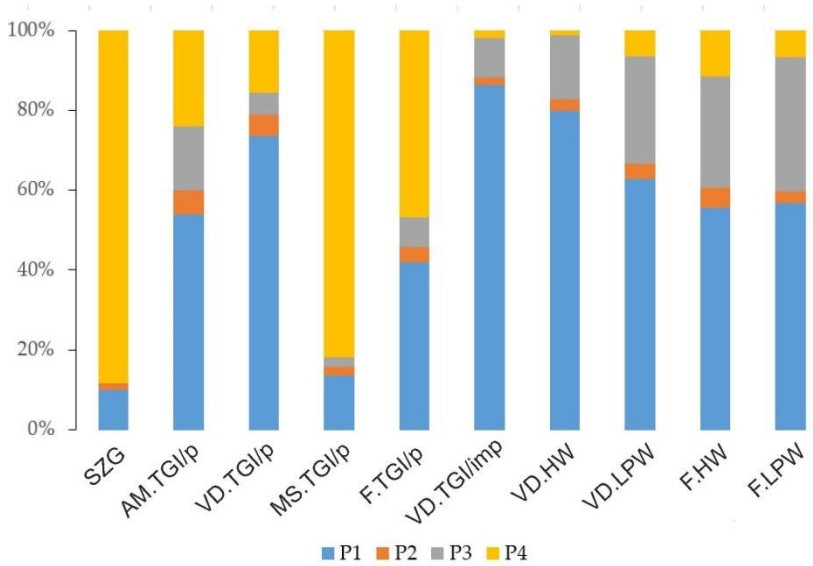

**Figure 6.** Relative abundance of the EEM PARAFAC compounds in various groups of ice samples.

A linear regression between the Box–Cox-transformed values of fluorescent DOM content (as the sum of P1–P4) and DOC concentration indicates a satisfactory fit between these parameters ($r^2$ = 0.83, *p* = 0.0000). In the plot (Figure 7), we observe a clear separation of the samples extremely enriched in both DOC and fluorescent DOM, including all the impure TGI (VD.TGI/imp) and the one Holocene IW sample (VD.HW). The highest data spread is indicated for the samples with DOC concentration below 10 mg/L, including glacial ice and pure TGI. There are two isolated groups of data points seen in the plot with higher and lower fDOM values in a common range of DOC content (marked as ovals in Figure 7). We suppose that samples located far below the regression line in the scatterplot (Figure 7), particularly SZ.G, MS.TGI/p, F.TGI/p, and two distinct samples of VD.LPW and VD.TGI/p, may contain a relatively higher proportion of non-fluorescent DOM contributing to DOC.

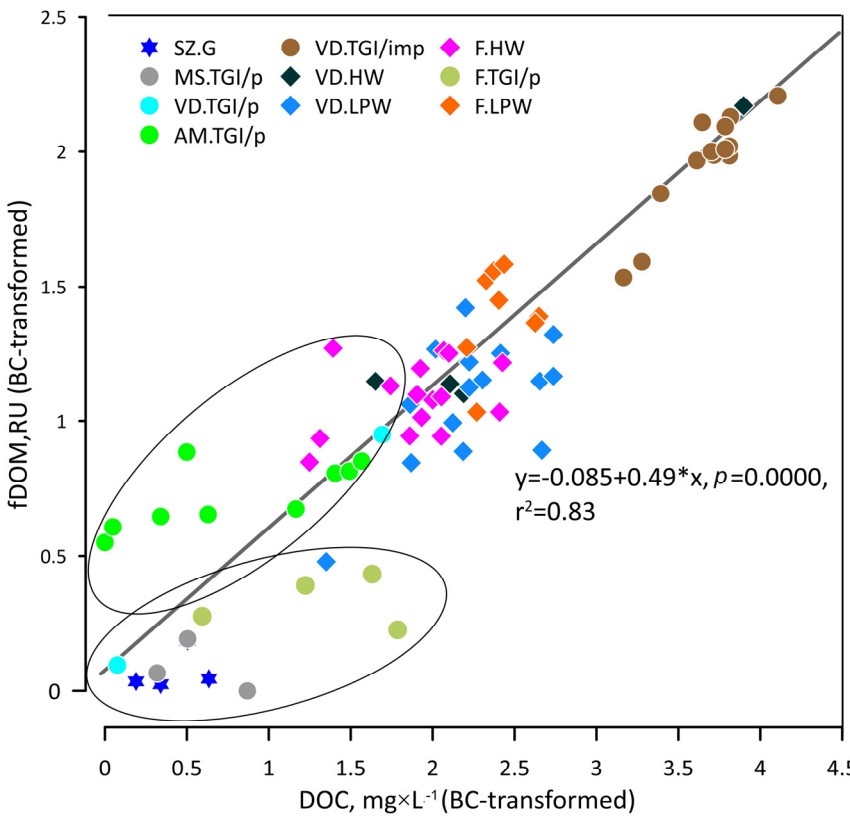

**Figure 7.** Fluorescent DOM (fDOM) content (as sum of all PARAFAC components) plotted vs. DOC concentration. The values are Box–Cox (BC) transformed.

Concerning the P1 component, each of the categories reveals a statistically significant variation (*p* < 0.005) from any other as shown by the Kruskal–Wallis ANOVA test. The sequence of decreasing P1 contents is impure TGI (TGI/imp) > ice wedge (IW) > pure TGI (TGI/p).

The P2 component is highly variable in the TGI/p and quite homogenous in the TGI/imp samples. There is a significant difference (*p* < 0.005) between TGI/imp and both TGI/p and IW. TGI/p and IW are not statistically distinct; however, TGI/p shows a higher average value than IW. The component P3 indicates the following sequence of decreasing values: TGI/p > IW > TGI/imp, identical to the one defined for P2. The TGI/p samples appear to be especially enriched in protein-like DOM, represented by the P4 component. The observed order of decreasing values of P4 is TGI/p > IW > TGI/imp with all the differences between the ice categories statistically affirmed by Kruskal–Wallis ANOVA. Distribution of the protein-like component P4, marking the fresh, autochthonous fDOM fraction, is shown in Figure 8D,E.

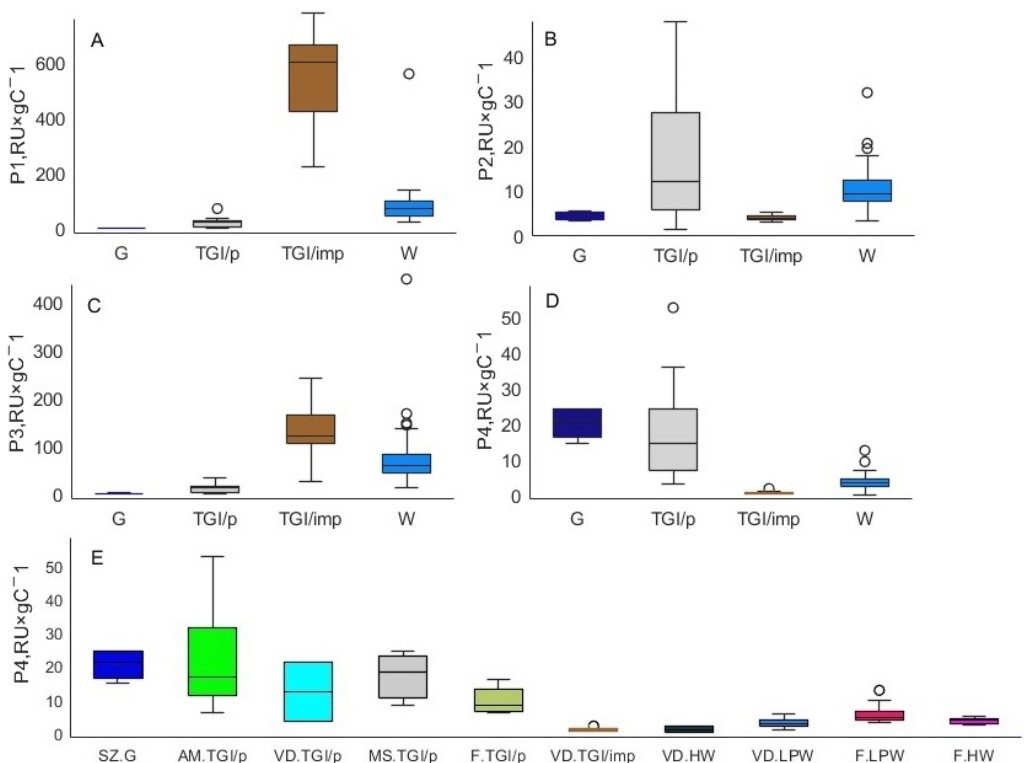

**Figure 8.** Box-plots of DOC-normalized values of the EEM PARAFAC components, P1 (**A**), P2 (**B**), P3 (**C**), P4 (**D**), in the categories of the ice samples, distribution of the DOC-normalized values of P4 component in the individual groups of ice samples (**E**).

*4.5. Results of Principal Component Analysis: The Variables' Interrelation and Data Classification*

The first two principal components explained 83.61% of the total variance in the dataset with 50.26% and 15.06% covered by PC1 and PC2, correspondingly. It can be seen from the PCA biplot in Figure 9 that all the variables positively correlate to each other as detected by their similarly negative loadings in respect of PC1, ranging from $-0.431$ ($SO_4^{2-}$) to $-0.93$ (TDS). Among the examined dissolved compounds, $SO_4^{2-}$ and $CH_4$ showed the least correlation with PC1 as indicated by loading values slightly below 0.5.

Relative to the PC2 axis, the variables fall into two groups characterized by negative and positive loadings, respectively. The group of variables with negative PC2 loadings includes DIC, DOC, $CO_2$, $CH_4$, EEM PARAFAC components P1–P3, and solid fraction. The PC2 loadings within this group vary from $-0.66$ ($CH_4$) to $-0.16$ (EEM PARAFAC component P3). We denote this group of variables as "carbon-dominated", because it contains all the bulk carbon cycle parameters examined in this work. The group with positive PC2 loadings includes $Na^+$, $Cl^-$, $SO_4^{2-}$, TDS, FI, EEM PARAFAC component P4, $K^+$, DIN, $Ca^{2+}$, and $Mg^{2+}$. The PC2 loadings within this group range from 0.14 ($Na^+$) to 0.69 ($Ca^{2+}$). We denote this group of variables as "salt-dominated", because it contains all the ionic compounds, except for $HCO_3^-$ and $CO_3^{2-}$, summarized as DIC. Within the "salt-dominated group" we observe a more or less pronounced relation in terms of PC2 loadings between $K^+$, $Na^+$, and $Cl^-$ as well as between $Ca^{2+}$ and $SO_4^{2-}$. $Cl^-$ as a major ion is closely related to TDS.

The humic-like DOM components P1, P2, P3 exhibit clear correlations among themselves, as well as to DOC and S (%). In contrast, the protein-like component P4, representing freshly derived autochthonous DOM, expresses no correlation to other EEM PARAFAC components nor to any of the bulk dissolved carbon species. However, it weakly or moderately correlates to $Mg^{2+}$, DIN, and FI. Generally, the protein-like component P4 is associated with a "salt-dominated" group of variables in the PCA biplot as opposed to the humic-like components P1–P3 which are closely linked to the "carbon-dominated" group.

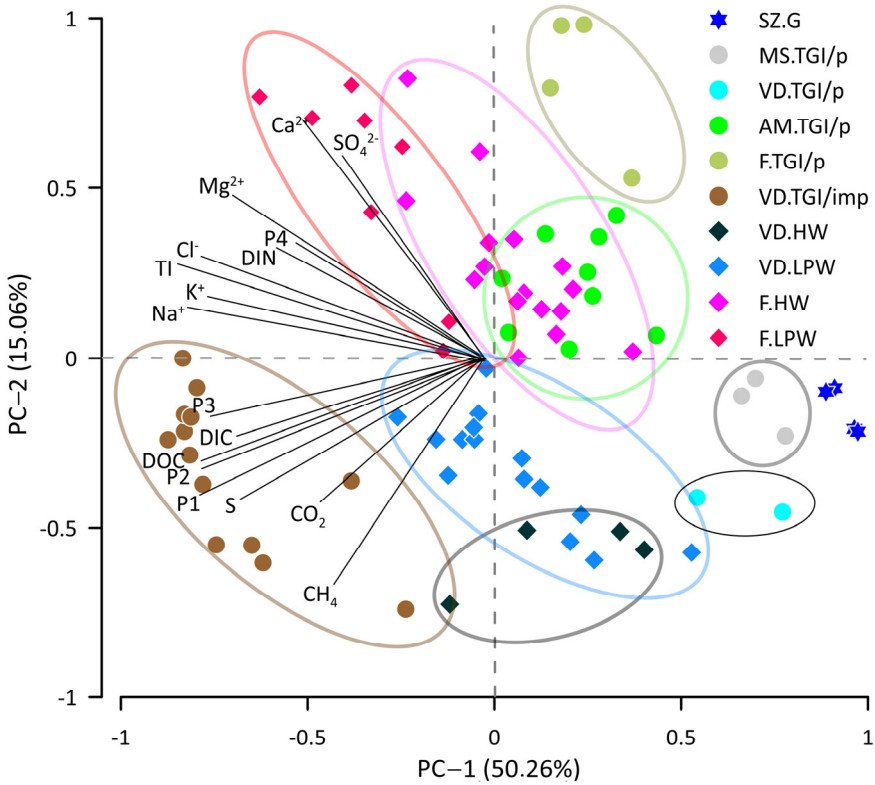

**Figure 9.** Biplot for the first two principal components, results of the PCA of the ground ice dataset. The ovals outline the data clouds of corresponding groups of samples.

The analyzed ice samples are widely scattered across the biplot. The glacier ice samples from Severnaya Zemlya (SZ.G), included in the sample set for comparison with ground ice, are remarkable for the highest positive loadings on PC1 and moderately negative values on PC2. The pure TGI samples from the Marre-Sale site in western Yamal (MS.TGI/p) appear to be quite similar to those of the glacier ice by the distribution of data points, indicating the highest positive loadings on PC1 among the analyzed ground ice samples.

The pure TGI sampled near Amderma, Yugorsky Peninsula (AM.TGI/p), is entirely located within the positive field of PC2 loadings, indicating lower values on PC1 and higher ones on PC2 compared to the previous group of TGI (MS.TGI/p). Ice wedges sampled from the Faddeevsky Peninsula (F.TGI/p) have a range of PC1 loadings similar to those of AM.TGI/p samples but considerably higher loadings on PC2 with the maximum values in the whole dataset. The last distinct group of TGI combines the samples from the Vaskiny Dachi site, central Yamal (VD.TGI/imp), being referred to as impure ice, containing >5% (42.01% on average) of the solid fraction (S, %). Those samples are most divergent from the rest of the data points and characterized by negative ranges of loadings on both PCs. VD.TGI/imp samples show minimum values on PC1 loadings in the dataset and vast scattering along the PC2 axis. PC1 and PC2 loadings for a sufficient part of the samples in this group are likely to be negatively correlated, suggesting the simultaneous drop of PC1 and growth of PC2 loading values. The single samples VD.TGI/imp 6 and VD.TGI/imp 10 are most deviant from the other samples of this group, where the first one is notable for the highest loading on PC1, but the second shows the highest value on PC2. Assuming the variable eigenvector position in the biplot, the variation within the VD.TGI/imp sample set depends mainly on parameters of the "carbon-dominated" group of variables (DIC, DOC, $CO_2$, $CH_4$, EEM PARAFAC components P1–P3, solid fraction content).

The Late Pleistocene IWs sampled from retrogressive thaw slump in the Vaskiny Dachi site, central Yamal (VD.LPW), are distinguished by highly variable values of PC1 loadings (from medium positive to medium negative) and exclusively negative loadings of PC2.

There is likely an inverse correlation between PC1 and PC2 loading values within this group as follows from the data cloud orientation in the biplot. Quite a similar distribution pattern was observed for the samples of VD.TGI/imp, but within the range of higher negative loadings on PC1. Four samples of Holocene IW (VD.HW) obtained in the Vaskiny Dachi site are partly mixed in the biplot with Late Pleistocene IW (VD.LPW), but they are distinguished by slightly lower values of PC2 loadings. The sample VD.HW 3, remarkable for an enormously high (for IW) solid fraction content (7.44%) and $NH_4^+$ concentration (5.15 mg/L), is highly divergent from other samples of the group and appears in the biplot close to impure TGI sample VD.TGI/imp 10 with the highest loading on PC1 in the VD.TGI/imp group. The IWs from the Faddeevsky Peninsula represent the last two groups of IWs in the dataset, both located in the biplot within the range of positive PC2 values, unlike the Yamal IW examined above. Holocene IWs (F.HW) show mostly positive values of PC1, except for the samples F.HW 5, F.HW 9, and F.HW 10, indicating the negative loadings on PC1 coupled to elevated (positive) loadings on PC2. All these samples are comparably enriched in major ions, especially in $SO_4^{2-}$, while having an average level of DOC, DIC, and fluorescent DOM components. The most of the Late Pleistocene IW samples from Faddeevsky (F.LPW) are characterized by the low (highly negative) loadings on PC1 and high (positive) loadings on PC2, excluding the samples F.LPW 2 and F.LPW 6, notably depleted in solid fraction content (S, %) as well as in dissolved ions. The Holocene and Late Pleistocene IWs from Faddeevsky (F.HW. and F.LPW) are well separated by the PCs loading distribution in the biplot.

## 5. Discussion

### 5.1. Variation of Basic Bulk Parameters of the Ground Ice as Indication of Their Biogeochemical Heterogeneity

Our results show that the solid matter content (S, %) in the ground ice samples is likely a basic and reliable marker of variations between the chosen sample categories: pure tabular ground ice (TGI/p), impure tabular ground ice (TGI/imp), and ice wedge (IW, W). Apart from cryogenic mechanisms eventually driving the formation of relatively pure and relatively impure ground ice bodies, we propose that the solid fraction content is one of the critical parameters determining both bulk and individual chemical composition of ground ice samples, including the DOM fractions and the coupled DOC pool heterogeneity. The box-plots of the S (%) and TI (%) show coinciding patterns of the median values' distribution (Figure 3A,B) which can preliminarily indicate the solid material contacting source water as a major source of the dissolved salts. In the case of the ice wedges located near the sea coast, particularly the samples from Faddeevsky, marine aerosols may contribute to the ion composition [30,47].

DOC concentrations in the analyzed samples are highly variable. The maximum for our dataset is encountered in the impure TGI (TGI/imp) with maximum and median values of 111.0 mg/L and 66.6 mg/L, respectively. Similar DOC loadings were reported for permafrost bogs, wetlands, and coastal tundra [48]. Our values of DOC concentrations in the IW samples are generally in accordance with those previously reported for Western Siberia [36], Eastern Siberia, and Alaska IWs [9]. We detected no significant variations in DOC content between the IWs from spatially remote geographical sections (Yamal and Yugorsky peninsulas). Moreover, the Late Pleistocene and Holocene IWs from Faddeevsky exhibit median values (10.76 and 7.61 mg/L, respectively) similar to those published for the Late Pleistocene and Holocene IWs of Eastern Siberia (11.1 and 7.3 mg/L, respectively) [9]. The median DIC concentration is again very high (16.73 mg/L) in the organic rich impure TGI from Yamal (Vaskiny Dachi site). The intensive processes of microbial respiration could be responsible for the DIC enrichment in the host sediments and source water, if the carbonate leaching by water streams is rather unlikely. In other sampled locations, the DIC values were comparable or lower than reported for ground ice of the Russian Arctic (the lowest is peculiar to pure TGI from the Marre-Sale site).

The examined distribution of the bulk parameters is likely to indicate that the local environment with distinctive features of OM transportation and mineralization plays a crucial role in DOC and DIC enrichment or depletion. The major variation in terms of these parameters is associated with the ground ice types, suggesting statistically distinct ranges of values. The minor intergroup variation may be also linked to the age of the IW.

*5.2. Carbon-Bearing Gas ($CH_4$ and $CO_2$) Concentrations Characterizing the Greenhouse Gas Storage and Various Conditions of the Ground Ice Formation*

Methane concentrations are extremely high in the impure TGI samples which is consistent with the published data on Western Siberia ground ice [32,36,49]. However, it is notable that the Yamal Late Pleistocene IWs are also extremely rich in methane. Although the median methane concentrations are 1.7 times higher in TGI (VD.TGI/imp), the IWs (VD.LPW) are not statistically different by $CH_4$ value distribution. This generally suggests the source water enrichment in dissolved or/and gaseous methane before its incorporation into forming IWs. Methane-rich host sediments are the most probable source of methane enrichment in Late Pleistocene IWs from Vaskiny Dachi, explaining the observed resemblance between IWs and TGI. At the same time, IWs from Faddeevsky (both Late Pleistocene and Holocene) reveal a more than 2000 times lower methane content, which may be due to both geographical factors, implying lower biological activity at higher latitudes, and the generally more favorable conditions for methanogenesis in the continental tundra biotopes (Vaskiny Dachi, central Yamal) than in other settings (Faddeevsky, New Siberian Islands).

The highest variability of methane content among all the measured and employed geochemical parameters is evidence of the sporadic nature of methane distribution with contrasting zones of enrichment in the ground ice bodies with different genetic features. The cryogenic processes driving the segregation of gas saturation zones in permafrost may explain this phenomenon for TGI, characterized by epigenetic freezing [50]. We also cannot rule out the possibility of methane formation under cryogenic conditions that may also contribute to the observed methane variability [51]. The thawing methane-rich ground ice deposits could be a considerable source of the predominantly ancient methane in the local ecosystems, however, the rate of this contribution in the net methane emission is difficult to predict due to heterogeneous methane distribution in the ground ice deposits.

Methane and carbon dioxide are partners in methane and carbon cycles, suggesting that methane is readily oxidized by microorganisms either in aerobic or in anaerobic conditions (utilizing widespread electron acceptors, such as $SO_4^{2-}$, $NO_3^{2-}$, etc.) [52]. In our dataset, when combining the TGI and IWs there is no correlation between $CH_4$ and $CO_2$ which could have explained their relationships within a common carbon reservoir. In this regard, our observations are consistent with the recent results published by [39], but in other works a negative correlation has been observed between $CO_2$ and $CH_4$ in the IWs [38,53]. Probably, there are many factors controlling the distribution and partitioning of the carbon-bearing gases, including labile organic matter respiration and acetolactic methanogenesis, yielding $CO_2$ and coinciding with its uptake for hydrogenotrophic methanogenesis.

Summarizing the above, we may conclude that for our dataset, central Yamal TGI, categorized as impure (TGI/imp), containing >10% (~40% on average) solid fraction (S, %), and Western Siberia IWs from the Vaskiny Dachi site are the most important potential sources of the "direct emission" of methane upon thawing. All the other groups and types of ground ice studied in this work seem to be negligible in this respect.

*5.3. Fluorescent DOM Composition and Biolabile DOM Fraction*

In this work, we examine the EEM PARAFAC components as the most reliable qualitative and quantitative indicators of the fluorescent DOM representing a significant part of all DOM [40]. The valid PARAFAC model for EEMs of a variety of ice samples (n = 216), including the various locations across the Russian Arctic, show the adequacy of the approach based on fluorescent DOM measurements. All the decomposed PARAFAC

components were recognized through the EEM PARAFAC library search, indicating the great resemblance to the components described in the literature [44–46]. The composition of the fluorophores detected in the present work fully coincides with those published in our previous work [30], which was carried out on 119 EEM samples, including a similar range of sample types. The finite PARAFAC dataset of 246 EEMs that was employed in the present work fully contains the previous database. Regarding the linkage of our PARAFAC components to the fluorophores found in other ice and water samples from the cryosphere, we should note the following. The humic-like DOM component P1, characterized by the longest excitation/emission (ex/em) wavelengths (370/470), is likely similar to KW5 (365/456) found in the Yedoma IW of northeast Siberia [18] and to C2 (365/473) detected in glacial ice of southwest Greenland [54]. P1 was predominant in the Yamal impure TGI and less abundant in the pure TGI (Figure 6). P2, with the shortest emission wavelength of a humic-like fluorophore (310/420), is likely to be related to KW4 (300/420) of the northeast Siberia IW [18] and C3 of the Greenland glacial ice [54]. P3, with the shortest excitation wavelength of humic-like DOM (260/470), resembles KW1 [18] and component 3 in the PARAFAC model comprising a great variety of glacier samples, from Canada to Antarctica [55]. The protein-like DOM component P4, classified as tryptophan-like, is rather close to component 3 detected in the glacier DOM [55]. These observations exhibit a spectral composition integrity of fluorescent DOM in the cryosphere, suggesting either a certain commonality of the DOM sources within the sampled environments or a convergence of various DOM species in their fluorescence features.

The median composition of the ground ice fluorescent DOM indicates that the autochthonous protein-like component P4 is predominant in the glacier ice and pure TGI (TGI/p) samples, characterized by lower DOC concentrations. The highest loadings of both DOC and fluorescent DOM are characteristic of the impure TGI (TGI/imp). P1, the longest wavelength fluorophore and probably the highest-molecular-weight humic-like DOM component among the identified fractions, is responsible for the observed DOC enrichment. The criterion of biolability is the most critical for a DOM composition assessment in view of biogeochemistry. Low molecular weight and reduced aromaticity are considered major signatures of a labile DOM vulnerable to rapid biological oxidation/mineralization [25]. Assuming that, all the measured humic-like compounds (P1–P3) are rather recalcitrant compared to the nitrogen-rich protein-like component P4, ultimately linked to local microbial biomass. P4 is not correlated to the humic-like components, indicating that autochthonous and allochthonous DOM fractions are likely to be disconnected in terms of the local biogeochemical cycling occurring in source water. Thus, the DOC-normalized values of P4 loadings can serve as a reliable tracer of the ground ice DOM biogeochemical "quality". However, as shown for Eastern Siberian Yedoma IWs, the component KW4, coinciding with our humic-like component P2 by the Ex/Em maximum, was positively correlated with DOC loss (biodegradable DOC (BDOC) amount in standardized aerobic incubation experiments [18]). Thus, based on the literature data, we propose that our component P2 may also represent a labile DOM fraction along with component P4 characterizing autochthonous protein-like DOM. In the box-plot (Figure 8A–D), the distribution of this parameter is indicated for the particular groups of ice samples. A statistically significant difference (Kruskal–Wallis multiple test) is observed between the biolabile DOM fraction of the pure TGI from Yugorsky Peninsula (AM.TGI/p) and impure TGI and Late Pleistocene IW from central Yamal (VD.TGI/imp and VD.LPW, respectively), as well as between VD.TGI/imp and Holocene IW from New Siberian Islands (F.HW) at $p < 0.005$. Finally, we observe the following sequence of decreasing values of the speculative biolabile DOM fraction in the analyzed ground ice samples: AM.TGI/p > VD.TGI/p > F.HW > F.LPW > F.TGI/p > VD.LPW > VD.TGI/imp.

Given the correlation between fluorescent DOM and DOC (Figure 7) demonstrated by $r^2 = 0.83$ ($p = 0.000$), we propose a possibility to use fluorescent DOM for ice DOC structure representation. We speculate that lower fDOM values at a given DOC concentration could indicate a higher contribution from non-fluorescent DOM, including organic

acids, carbohydrates, water soluble lipids, etc. [56]. In this view, the AM.TGI/p samples are supposed to be relatively depleted in non-fluorescent DOM. Non-fluorescent DOM, represented predominantly by low-molecular-weight components, is likely to indicate the labile DOM fraction along with protein-like components of the fluorescent DOM. Thus, the quantification of the bulk non-fluorescent DOM, involving the regression between fDOM and DOC, is a promising tool, however, it should be metrologically assessed.

Summarizing the above, it can be argued that the ground ice samples from a great variety of settings contain high-molecular-weight humic-like DOM as a major recalcitrant fraction of DOM (components P1, P3) and a low-molecular-weight humic-like component (P2) plus an autochthonous protein-like component (P4) as minor labile fractions of DOM. Therefore, the largest pool of DOC is associated with DOM, requiring intensive enzymatic processing for generating substrates consumable by chemoorganotrophic microbiota, operating aerobically and with respiration coupled to $CO_2$ and $CH_4$ production in aerobic and anaerobic conditions, respectively. The enzymatic processing, involving the cleavage of high-molecular-weight compounds with increased aromaticity into low-molecular-weight aliphatic species, is the most energy-consuming stage of this catabolic pathway [18,19]. The excretion of the hydrolytic enzymes is coupled to significant carbon loss by the operating microbial biomass.

Probably, the role of the labile permafrost-derived carbon is in the so-called priming effect on mineralization of the surrounding soil carbon [20]. Relative to ground ice, this was shown by published results of experiments on the addition of IW thaw water, relatively enriched in ancient labile carbon, to modern permafrost soil leachates, relatively depleted in labile forms, that eventually led to increased carbon loss in the experimental solutions [18]. Low solid fraction content in ground ice (except the impure TGI) may contribute upon their proposed thawing to the increase in water connectivity, providing effective transport within a disturbed zone of the low-molecular-weight DOM, highly vulnerable in terms of microbial uptake and respiration. The combination of our current descriptive data on the ground and glacier ice geochemical composition with the further incubation tests, involving the published recommendations [18,19,57], will allow us to obtain reliable data, which can be used for spatial generalization of the ground ice thawing influence on modern Arctic ecosystems in terms of the carbon cycle.

*5.4. Variations in the Ground Ice Samples' Geochemical Composition and Their Possible Drivers Explained by Exploratory Statistical Analysis*

The results of PCA (Figure 9) define the pronounced enrichment of the ice samples in dissolved compounds within a range between the glacial ice and impure TGI as the most pure and "contaminated" species of our dataset, respectively. This trend is responsible for 50.26% of the total variance in the database as shown by PC1. The minor, but representative, part of the total variance (15.06%) explained by PC2 is remarkable for a split in two groups: "carbon-dominated", which mostly drives the separation between different ice types, and "salt-dominated", which indicates the difference between the sites with similar ice types.

The IWs of northwest Canada also showed a common driver for the solute enrichment, expressed as high positive correlation of the measured ions with the first principal component [47]. However, for those datasets ultimately comprising the IWs, marine aerosol transport was postulated as a predominant factor of solute enrichment and, in our case, including both IWs and TGI, this is probably the contact with host sediments. However, this does not exclude marine aerosols' influence on the formation of nearshore Holocene IWs (Faddeevsky).

In spite of the definitely divergent origin, the pure TGI samples from the Marre-Sale site (MS.TGI/p) appear to be convergently close to the glacier ice samples (SZ.G) in terms of the geochemical parameter distribution. Therefore, the corresponding ground ice bodies/layers are likely to contain virtually no solid inclusions and be very poor in the dissolved species analyzed (ions, DOC, gases, etc.). The pure TGI samples from the Vaskiny Dachi site (VD.TGI/p) are distinguished from those of Marre-Sale (MS.TGI/p)

by a combination of low ion and DOC content with relatively elevated methane level (responsible for lower values of PC2). The pure TGI from the Amderma site (AM.TGI/p) is mixed with ice wedges from Faddeevsky in the PCA biplot, suggesting the geochemical convergence of the genetically different ice samples from spatially remote sites. However, there is a clear separation between Yamal and Faddeevsky IWs mostly based on ion composition, which is probably linked to variability in the ice formation conditions along the geographical scale or due to the spatial horizontal variation in the ice formation environments. The outstanding abundance of methane in the Yamal IWs relative to the IWs of Faddeevsky clearly shows a better condition for methanogenesis and methane migration in corresponding environments, probably associated with more favorable conditions for basic biological productivity. The Holocene IW of the Yamal Peninsula (VD.HW) shows a slightly different pattern of PC distribution due to higher DOC and lower mineral content. The Holocene and Late Pleistocene IWs from Faddeevsky are also partly separated along the PC1 axis, with distinct groups of samples highly correlated to the $SO_4^{2-}$ eigenvector due to the enrichment in not of sea-salt (nss) origin [30]. Impure TGI and Late Pleistocene IWs from Yamal reveal the common patterns of distribution along the PC2 axis, which may be associated with the IW formation in the TGI body eventually responsible for their characteristic enrichment of methane and humic DOM.

Humic-like DOM including the PARAFAC components P1–P3 obviously hosts the larger pool of DOC ultimately sourced from terrigenous higher plant debris. Gas compounds ($CH_4$ and $CO_2$) do not significantly correlate to any of the parameters, revealing a weak positive correlation to each other as well as to DOC and humic-like DOM. This can be explained by specific features of in situ methane generation and transportation, which are totally different for IW and TGI. $CO_2$, except for the equilibrium atmospheric constituent of the ice-captured air bubbles, is mainly associated with biological oxidation of OM and $CH_4$ prior to freezing or under subsequent cryogenic conditions. Both DIC and $CO_2$ correlate weakly or moderately with DOM components P1–P3 but do not correlate with the most biolabile P4 component (protein-like DOM). Thus, the autochthonous DOM does not predict the intensity of OM mineralization in the ground ice source water, probably due to its minor contribution in C pool volumes. However, the refractory, but more abundant, humic-like components P1 and P2 are likely to be more quantitatively associated with both carbon pools: oxidized and reduced. Anyway, the interrelations between the individual parameters of the ground ice geochemical composition are very complicated and influenced by multiple factors related to the reaction and transportation processes operating within a long timescale.

## 6. Conclusions

This study has revealed a variability of glacial and ground ice samples from various geographic locations when assessed through various geochemical parameters.

While material might be transported into the source water by various means, such as aerosol transport (particularly in the case of ice wedges) [58–60], it appears that the interaction with solid material—such as sediments, detritus, and vegetation—is likely the overriding process in the ground ice enrichment in all the analyzed dissolved components. This is consistent with previous studies indicating that terrestrial leaching is a major source of DOC [9]. We analyzed the distribution of a number of geochemical parameters relevant to the carbon cycle and carbon storage in ground ice.

Methane exhibits sporadic and contrasting distributions in the ground ice, with Yamal ground ice samples consistently showing higher levels of methane compared to those from other geographical areas. Notably, methane concentrations are similarly high in both impure tabular ground ice (TGI) and Late Pleistocene ice wedges (IWs), up to 14,138.08 ppmV.

The highest dissolved organic carbon (DOC) concentration (up to 111 mg/L) was detected in the impure tabular ground ice from Yamal which has extremely high solid fraction content (>50%). A lower DOC content was encountered in the ice wedges and the lowest in the pure TGI, which was even lower than in the glacier ice.

Excitation–emission matrix (EEM) fluorescence coupled to parallel factor analysis (PARAFAC) has demonstrated the substantial variability of dissolved organic matter in different ice types. The sum of the fluorophores deconvoluted by PARAFAC strongly correlates to DOC, which proves the potential of using fluorescent DOM components for differentiation of bulk DOC into fractions with various origins and biogeochemical behaviors. The protein-like component, indicating an autochthonous source, showed no correlation to humic components and is proposed to be associated with microbial abundance in the ice [61]. The pure tabular ground ice samples exhibit the highest rate of fresh easily degradable DOM in the bulk DOC, which may be responsible for the amplification of permafrost organic matter decomposition upon thawing.

The geochemical variable composition allows us to discern between Late Pleistocene and Holocene ice wedges from the Faddeevsky Peninsula, as well as to differentiate the Yamal ground ice from ice of other locations. However, there is a convergence in geochemical composition between the glacial ice samples from Bolshevik Island and pure TGI of Yamal Peninsula.

The obtained results represent valuable reference information on quantitative parameters of the carbon cycle in ground ice, providing geochemical classification of the samples with the possibility to find hidden genetic links between the ice types. The validated model of the fluorescent DOM composition allows for insight into ground-ice-derived DOC heterogeneity and DOM "biogeochemical quality".

We consider that further study should combine the current results with data of DOC experimental biodegradation performed on representative samples. This will certainly clarify the question on DOM lability and potential of the ground ice DOM to generate greenhouse gases upon putative mobilization into the modern cycle with climate change.

**Supplementary Materials:** The following supporting information can be downloaded at: https://www.mdpi.com/article/10.3390/geosciences14030077/s1, Table S1: Gases and bulk biogeochemical components in ice samples; Table S2: Ion concentrations in the ice samples; Table S3: EEM-DOM PARAFAC component loadings; Table S4: Pearson correlation coefficient matrix of the EEM-DOM PARAFAC components.

**Author Contributions:** Conceptualization, P.S. and A.P.; methodology, P.S., A.K., E.S., S.M., N.B., A.L. and I.T.; software, W.H. and P.S.; writing—original draft preparation, P.S., A.P., A.K., E.S., N.B. and P.G.; writing—review and editing; visualization, P.S., P.G. and A.P.; supervision, I.S. and M.L. All authors have read and agreed to the published version of the manuscript.

**Funding:** This research was supported by the Russian Science Foundation, grant number 23-27-00123.

**Data Availability Statement:** Data are contained within the article and Supplementary Materials.

**Acknowledgments:** The authors would like to thank Victor Bogin (VNIIOkeangeologia) for the great assistance in the ground ice sample processing and the anonymous reviewers for the detailed reports that allowed significant improvement of the manuscript. Additionally, we thank the staff of the Earth Cryosphere Institute for their assistance in the field work.

**Conflicts of Interest:** The authors declare no conflicts of interest.

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
