# Peer review of "Characterizing Dissolved Organic Matter and Other Water-Soluble Compounds in Ground Ice of the Russian Arctic: A Focus on Ground Ice Classification within the Carbon Cycle Context"

_geosciences, doi:10.3390/geosciences14030077_

Round 1

Reviewer 1 Report

Comments and Suggestions for Authors

This paper focuses on studying dissolved organic matter (DOM) in Arctic ground ice. It examines the geochemical characteristics of ground and glacier ice across various Arctic locations, analyzing their ion composition, carbon-bearing gases, bulk biogeochemical indicators, and fluorescent DOM fractions. The study highlights the interaction between solid materials and ground ice in enriching dissolved compounds. It finds a predominance of terrigenous humic-like DOM in most ice samples. It notes a strong correlation between fluorescent DOM components and dissolved organic carbon (DOC), indicating the adequacy of the estimation method. The research underscores the importance of ground ice in the carbon cycle, particularly in the context of climate change and permafrost thawing.

I am not an expert in biogeochemistry, which limits my ability to fully assess the methods used. I suggest listing more recent work related to GHG emissions from thawing permafrost (e.g., McGuire et al., 2018 and Natali et al., 2019). This study has significant implications for understanding the carbon cycle and climate change, especially regarding greenhouse gas emissions from thawing permafrost. The technical nature of the paper might be challenging for readers without a background in geochemistry or Arctic studies. While the study is detailed in its specific field, it could benefit from a broader discussion on the global impact of its findings.

McGuire, A. D., et al., (2018), The Dependence of the Evolution of Carbon Dynamics in the Northern Permafrost Region on the Trajectory of Climate Change. DOI:

Natali, S.M., Watts, J.D., Rogers, B.M., et al. Large loss of CO2 in winter observed across the northern permafrost region. Nat. Clim. Chang. 9, 852–857 (2019). https://doi.org/10.1038/s41558-019-0592-8

Author Response

Dear Referee,

Thank you very much for your comments on our manuscript.  It was our omission to leave the indicated state-of-art research works without consideration. They concern global climate change projections involving permafrost area carbon dynamics under various conditions (mitigations etc.). The phenomenon of the increased carbon loss in the high Arctic during winter exceeding the vegetation cycle is very spectacular. Unfortunately, the topic of our manuscript, although concerning biogeochemistry, is still far away from the carbon cycle modeling and climate change projections. In this work we study geochemical variability of the ice samples and composition of their dissolved organic matter. Probably, fluorescent dissolved organic matter composition quantification using EEM PARAFAC analysis would help in defining the labile and recalcitrant fractions of organic matter in biogeochemical modeling coupled to experimental incubation. We plan experimental work on selected samples from our dataset with simultaneous measurements of key parameters of carbon cycle: CO2, DIC, DOC (biodegradable DOC) and DOM fractions. This will allow us to trace the transformation of DOM into CO2. In this context the GHG emission modeling performed by Knoblauch et al. Methane production as key to the greenhouse gas budget of thawing permafrost. Nat. Clim. Change, 2018, 8, 309–312. is closer to our findings than global projecting by McGuire, A. D., et al., (2018). Anyway in order to improve the Conclusions (which was indicated as a point of estimation) we totally reworked the text of this section:

 Conclusions

This study has revealed a variability of ice samples from various geographic locations when assessed through various geochemical parameters.

While material might be transported into the source water by various means, such as aerosol transport (particularly in the case of ice wedges) [59-61], it appears that the inter-action with solid material—such as sediments, detritus, and vegetation—is likely the overriding process in the ground ice enrichment in all the analyzed dissolved components. This is consistent with former studies indicating the terrestrial leaching is a major source of DOC [9]. We analyzed the distribution of a number of geochemical parameters relevant to carbon cycle, carbon storage in the ground ice.

Methane exhibits sporadic and contrasting distributions in the ground ice, Yamal ground ice samples consistently show higher levels of methane compared to those from other geographical areas. Notably, methane concentrations are similarly high in both impure tabular ground ice (TGI) and Late Pleistocene ice wedges (IW), up to 14138.08 ppmV.

The highest dissolved organic carbon (DOC) concentration (up to 111 mg/L) was detected in the impure tabular ground ice from Yamal with extremely high solid fraction content (> 50%), the lower DOC content was encountered in the ice wedges and the lowest in the pure TGI, which was even lower than in the glacier ice.

Excitation-Emission Matrix (EEM) fluorescence coupled to Parallel Factor Analysis (PARAFAC) have demonstrated the substantial variability of dissolved organic matter in different ice types. The sum of the fluorophores deconvoluted by PARAFAC strongly correlates to DOC, which proves the potential; of using fluorescent DOM components for differentiation of bulk DOC into fractions with various origin and biogeochemical behavior The protein-like component, indicating an autochthonous source, showed no correlation to humic components and is proposed to be associated with microbial abundance in the ground ice [62]. The pure tabular ground ice samples exhibit a highest rate of fresh easily degradable DOM in the bulk DOC which may be responsible for the amplification of permafrost organic matter decomposition upon thawing.

The geochemical variables composition allows us to discern between Late Pleistocene and Holocene ice wedges from the Faddeevsky Peninsula, as well as to differentiate the Yamal ground ice from ground ice of other locations. However, there is a convergence in geochemical composition between the glacial ice samples from Bolshevik Island and pure TGI of Yamal Peninsula.

The obtained results represent a valuable reference information on quantitative parameters of a carbon cycle in the ground ice, provide geochemical classification of the samples with possibility to find out a hidden genetic links between the ice types. The validated model of the fluorescent DOM composition allows for insight into a ground ice-derived DOC heterogeneity and DOM “biogeochemical quality”.

We consider that further study should combine the current results with data of DOC experimental biodegradation performed on representative samples. This will certainly clarify the question on DOM lability and potential of the ground ice DOM to generate greenhouse gases upon putative mobilization into modern cycle with climate change.

Reviewer 2 Report

Comments and Suggestions for Authors

In the submitted manuscript, the authors present ion and DOC quantification/characterization data from 4 different ice types at 5 different field sites in the Russian Arctic. They include detailed discussion of solids content, total and specific ion composition, DOC concentration and characterization using fluorescence methods, and perform principle component analysis of their ice samples. The data set is very valuable and represents a significant effort. I believe the paper and the dataset are ultimately worthy of publication and I congratulate the authors on a worthy study of what I imagine are difficult locations to sample. However, the presentation and discussion of the data need some significant modifications before publication.

The writing overall is pretty good, but the authors sometime use language that is vague. Simplifying some of the language may help readability. I provide some examples in the line-by-line comments below, but please do another read-through to clean up some of the language.

For site names, please use the same name or acronym throughout the manuscript. The authors sometimes switch between them (i.e. VD vs. Vaskiny Dachi vs. Yamal). For readers unfamiliar with the site locations, that consistency will help us follow the discussion.

I think the data would be better presented if the figures were condensed. Currently, multiple figures are shown for individual types of data (clustered by ice type, then by site+type, then by site+type with outliers removed). This could all be condensed into one very readable figures by plotting boxplots of each site+type, clustered along the x-axis by ice type, and colored by site type (preferably to match the PCA plot). If the authors use a log scale on the y-axis, that will make the difference between all samples much more readable. This will drastically simplify things for the reader and will allow the authors to present the site+type info for all the constituents.

Make sure to place the figures in the order that they are called out in the text. In particular, Figure 2 call-outs start with 2B, then 2C, then 2A, then 2F.

The glacier ice data should be included in the boxplots, and even if the values are much lower, using a low scale will highlight the trends.

For the linear regression analysis, this is an interesting idea, but the implementation and discussion around it could be improved. As it currently stands, the linear regression is driven largely by the impure VD.TGI samples. By log-transforming both axes, you will get a regression that better represents the lower end of the curve. As the discussion currently stands, there is little value added from this analysis. If this is included, please add some discussion of the relationship of various groups to the regression line. Points well below the regression line might be expected to have higher fractions of labile, non-fluorescent DOM. Comparing this to the data shown in Figure 14 would be valuable.

Regarding the PCA: The authors mention that the first component represents 50% of the total variability, while the second component represents only 15%, but I think it's important to revisit this in the discussion. Related to this, it could be made more clear that the DOC-related variables strongly drive the separation between different ice types, while the "salt-dominated" variables drive the difference between sites with similar ice types. This point is a little lost in the more nuanced discussion that the authors have. Important to note that the glacier samples are similar in salt content to the TGI samples. This is why it would be helpful to have the glacier samples in the earlier boxplots. Also regarding the PCA, there's a lot of discussion about the samples along the principle components, but given the clear trends with the variables, and the fact that at the end of the day the principle components are arbitrary, it would easier to follow if the separation of the points were presented more in their variation along the variable vectors.

Make sure to properly format subscripts and superscripts (I'm not sure if this is due to the submission method; if so, please ignore this comment).

Line by line comments:

Line 25: “release of large volumes of organic carbon” makes it sound like the carbon is going into the atmosphere as organic carbon (some is, as methane, but some is CO2).

Line 26-28: “Ground ice constitutes a considerable volume of the cryogenically sequestered labile dissolved organic carbon (DOC) subjected to fast mineralization upon thawing.” This is confusing and mixing units a bit. Maybe say "Ground ice contains a considerable mass of ...".

Line 38: “indicating the adequacy of the estimation.” What estimation are the authors referring to here? This language is too vague.

Line 39: “exhibit the highest biogeochemical quality”. Referring to the biogeochemical “quality” is too vague, please be more specific. Even something like: exhibit the highest concentrations of ions/redox-active species/DOC (whichever one the authors are referring to).

Line 56: “ground ice-bond carbon reservoir”. I’m assuming the authors meant “ice-bound” here, but this is a bit confusing. Perhaps more clear to just say the “carbon reservoir frozen in ground ice”.

Lines 57-58: “pre-formed” and “pre-aged” are unnecessary here. What is pre-aged carbon? If including this language, please define or cite.

Lines 75-78: Long-winded sentence, rather confusing.

Line 79: IW is not defined yet.

Lines 66-104: There are good ideas and citations in here, however, the writing is too vague. Clean up and simplify this paragraph.

Line 146. Set up the photos in FIgure 2 in order that they are discussed for improved readability.

Line 356: "... in the ground ice samples categories make up a row". Would be clearer to say "in the ground ice samples are in the following order"

Line 371: I don't see the single outlier value on Fig. 8A for VD.HW. Labeling the outliners in the figure that you call out in the text will help the reader, since we don't have the site names for these.

Line 382: Variation in methane is shown in Fig. 8.

Line 383: VD.LPW was also excluded from 8A.

Line 404: The Pearson correlation coefficients are not shown in Table 2.

Line 407: "The relative abundances of the EEM PARAFAC compounds (%) based on their median concentrations (RU) clearly indicate the character of the fluorescent DOM composition intergroup variability". This is an example of the wording being overly wordy to the point of being hard to follow. Do you mean something like "Fluorescent DOM composition varied between groups as shown by the relative abundances of the EEM PARAFAC compounds"?

Line 516: "Assuming the variables eigenvectors position on the biplot, the variation within the VD.TGI imp sample set depends mainly on parameters of the “carbon-dominated” group of variables (DIC, DOC, CO2, CH4, EEM PARAFAC components P1-P3, solid fraction content)." The impure samples are separated from the rest of the samples by the carbon-dominated variables, but in terms of intra-group variation, they tend to vary parallel to the salt-dominated variables, particularly the Ca2+ and SO42-.

Line 550: I thinking bringing in the similarity between VD.HW3 and the impure samples (discussed on line 530) would help support the carbon content of the water is primarily driven by solids content.

Line 622: Please include some discussion of how prevalent each of these types of ground ice is, as importance of thawing is tied to both concentration of methane as well as total global volume of each type of ice. If that prevalence is not known, just add that caveat.

Line 623: Regarding the discussion of labile carbon, the P2 and P4 components are not going to be the most labile carbon substrates. I recommend bringing in additional information here about the discrepancy between the samples and the regression line (figure 11) as well as other metrics like SUVA254 (very low SUVA may point to a lot of LMW carbon) that may indicate the presence of labile C that wouldn't show up on an EEMs. This is partially addressed in line 656, but it's unclear about how the intercept would relate to the labile DOC, since the labile DOC will not be consistent across all samples.

Line 629: "approve the adequacy of the used approach". Please clarify wordings like this.

Line 671: Just because P4 is not correlated to the humics doesn't necessarily mean that it's reliable tracer of labile carbon. This is overstating the analysis. It's still a valid analysis to conduct, but you can't say that it's a tracer of all labile carbon.

Line 676: Please split out the two components (P2 and P4) when you plot them. They represent very different carbon fractions, and it would be helpful to the reader to see how each individually varies.

Line 697-711: this is potentially very interesting discussion but very hard to follow. Please elaborate on these points, add some more citations to support your hypotheses, and make it clear what data is from this study, what data is cited, and what you are hypothesizing.

Line 737: Is the variability linked to the formation conditions or the ion content of the source water?

Line 771: citation needed.

Line 801: Do the authors mean the holocene ice wedges of Faddeevsky? That's what the data indicates.

Line 802: How does this method reconstruct freezing conditions? That is not mentioned elsewhere in the manuscript. Seems like much of this could be linked to source water rather than climatic conditions. Additionally how does this fit into data classification for carbon cycle modeling? If the authors want to include statements like this they need to provide additional support.

Comments on the Quality of English Language

Overall the English is quite good. My only comment with the writing is to make some sentences more clear, often times the wording is very vague, to the point that the reader cannot determine the nuanced meaning of the statement.

Author Response

Dear Referee!

thank you very much for your detailed and attantive review, which will certainly improve a quality of our article if it is published. In the file attached you will find point-by-point answers to your comments accomponied by the revised  fragments of the text

Round 2

Reviewer 2 Report

Comments and Suggestions for Authors

Overall the manuscript is substantially improved, particularly with respect to the labile carbon discussion.

The primary issue is with the figures and figure references. Figure call outs are sometimes missed (Figs 3 and 4 don't have all the components of the figure referenced in the text). There are incorrect figure references on Lines 139 and 142. There is also substantial redundancy in the figures. Figures 3 and 4 show the same data, just grouped differently (Fig 8C is also a repeat of 8D) There are ways these could be presented on the same graph that would be far more clear to the readers. If the authors really want to keep them separate, it should be made clear in the text that the data is the same, they're just grouped differently. Figure 4B and 4C show the exact same data, just with two groups excluded from 4C (text describes it as one, but 4C also excludes Holocene IWs from Vaskiny Dachi). This could be circumvented by using a log scale on the y-axis of Fig. 4B as I mentioned in my first review. A log scale would be helpful for many of the plots, since right now the data is clustered near the bottom and difficult to make out any differences. Fig 7 is also redundant between A and B. I had recommended the log-log transformations, but for the regression as well as the visualization. Ultimately the authors should select the best data transformation (or no transformation) for their data and go with that. Again, if both presentations are necessary the authors should make sure to explain why they chose to present it in both ways.

I would also recommend a careful readthrough of the manuscript for grammatical errors and missing punctuation as there still many issues with that throughout the document.

I still think the authors focus too much on the PCA loadings rather than the inter- and intra-group distribution of the samples relative to the eigenvectors of the geochemical variables, but this is a common way to discuss these plots and the plot is there for the readers to interpret themselves.

If the authors address the concerns with the figures the manuscript is ready for publication.

Comments on the Quality of English Language

The English is good. Errors in the manuscript do not seem to be an issue of language barrier but simply need proofreading.

Author Response

Dear Referee, we provide our answers to the comments of the current revision in the attached file (green highlights).
